# Regulation of miR-181a expression in T cell aging

Zhongde Ye[1,2], Guangjin Li[1,2], Chulwoo Kim[1,2], Bin Hu [1,2], Rohit R. Jadhav [1,2], Cornelia M. Weyand [1,2] & Jörg J. Goronzy [1,2]

MicroRNAs have emerged as key regulators in T cell development, activation, and differentiation, with miR-181a having a prominent function. By targeting several signaling pathways, miR-181a is an important rheostat controlling T cell receptor (TCR) activation thresholds in thymic selection as well as peripheral T cell responses. A decline in miR-181a expression, due to reduced transcription of pri-miR-181a, accounts for T cell activation defects that occur with older age. Here we examine the transcriptional regulation of miR-181a expression and find a putative *pri-miR-181a* enhancer around position 198,904,300 on chromosome 1, which is regulated by a transcription factor complex including YY1. The decline in miR-181a expression correlates with reduced transcription of YY1 in older individuals. Partial silencing of YY1 in T cells from young individuals reproduces the signaling defects seen in older T cells. In conclusion, YY1 controls TCR signaling by upregulating miR-181a and dampening negative feedback loops mediated by miR-181a targets.

[1] From the Department of Medicine, Division of Immunology and Rheumatology, Stanford University, Stanford, CA 94305, USA. [2] Department of Medicine, Veterans Affairs Palo Alto Health Care System, Palo Alto, CA 94306, USA. Correspondence and requests for materials should be addressed to J.J.G. (email: jgoronzy@stanford.edu)

With the globally changing age demographics, age-associated morbidities have become a worldwide societal challenge and new approaches are needed to improve healthy aging. Aging of the immune system is one of the limiting factors, essentially affecting all organ systems[1,2]. The aging immune system is more inclined to elicit nonspecific inflammation, which accelerates degenerative diseases, notably seen in cardiovascular and neurodegenerative disorders[3–5]. Equally important, the decline in immune competence contributes to the increased morbidity and mortality from infections[6,7]. Vaccination holds the promise of a cost-effective intervention; however, vaccine responses are generally poor in the elderly and at best ameliorate disease. Even for recall responses with high doses of live attenuated varicella zoster virus (14× higher than the childhood vaccine), protection rates decline from 70% in the 50–59 years old to <50% in the young–old (60–75 years) and <30% in the old–old (>75 years)[7,8]. While annual vaccinations with the trivalent or quadrivalent influenza vaccine are recommended, the vaccine response is also unsatisfactory[9–11]. One major objective of immune aging research therefore is to identify defects in adaptive immune responses that impair the generation of immune memory and that can be successfully targeted[12].

A decline in the ability to generate new T and B lymphocytes with age and a failure in maintaining homeostasis in this intricate cellular system composed of naïve, memory, and effector cells of highly variable clonal sizes and a vast array of antigen receptors has been frequently suspected as an underlying cause of defective T cell immunity. However, recent studies have suggested that the homeostatic mechanisms for the CD4 T cell compartment are surprisingly robust, at least in healthy elderly. In spite of lacking thymic activity, the size of the compartment of circulating naïve CD4 T cells only moderately shrinks and the diversity of the T cell receptor (TCR) repertoire, while somewhat contracted, is still immense[13–15]. In fact, uneven homeostatic proliferation appears to be a greater threat to diversity than stalled thymic T cell production[16]. Defective vaccine responses therefore appear to be more related to impaired T cell function than numbers and diversity[17]. However, a single dominant functional defect, such as cellular senescence has not been found, and the overriding aging signature in cell biological studies of naïve and also central memory T cells from older individuals is dominated by markers of accelerated differentiation[18]. This is particularly evident in epigenetic studies of CD8 T cells from older individuals with chromatin accessibility maps of naïve CD8 T cells shifted to those of central memory CD8 T cells[19]. This epigenetic signature is only in part due to the accumulated memory CD8 T cells that assume a naïve phenotype[20–22]. A similar shift towards more differentiated state with age is also seen for central memory cells that exhibit features of effector T cells[19]. Moreover, terminally differentiated CD45RA effector T cells accumulate that have features of innate effector cells[23–25].

MicroRNAs are known to be an important driver of differentiation. Because they concomitantly reduce expression of many target molecules, their concerted activity may have a major influence although the effect size on each of this molecules is small[26,27]. We have previously hypothesized that changes in the age-associated expression of microRNAs targeting signaling pathways lead to defects that are seen with T cell aging. Based on our initial findings that naïve CD4 T cells from older individuals have reduced extracellular signal-related kinase (ERK) phosphorylation upon TCR stimulation, we have focused on miR-181a. miR-181a is highly expressed in T cells and is dynamically regulated during activation and differentiation[28]. Functionally, it is an intrinsic regulator of the TCR activation threshold by controlling the expression of the cytoplasmic DUSP6 and other

negative feedback pathways including PTPN22, SHP2, DUSP5, and SIRT1[29–31]. In addition, a reduced expression of PTEN and associated effects on the AKT-mTORC1 pathways have been described in one, although not confirmed in a second miR-181a/b1-deficient mouse[32,33]. miR-181a is highly expressed in thymocytes facilitating positive selection, but less so in single-positive thymocytes and peripheral T cells, where expression further declines with differentiation or TCR-mediated activation[31]. Expression in naïve CD4 T cells declines with age resulting in the overexpression of DUSP6 dephosphorylating pERK[34].

In the current study, we examine the mechanisms underlying the loss of miR-181a expression with age and T cell differentiation. Here we describe that the transcription of *pri-miR-181a/b1* is controlled by an enhancer region that is regulated by the transcription factor (TF) YY1. The decline in YY1 expression with age results in reduced miR-181a expression and concomitantly increased expression of DUSP6 and SIRT1 and the corresponding loss in TCR sensitivity to respond to stimulation.

## Results

**Age-dependent regulation of pri-miR-181a/b1 transcription**. In our previous studies, we had found a decline in miR-181a in naïve and, to a lesser extent, memory CD4 T cells with age impairing TCR sensitivity to respond to antigen stimulation[34]. To determine whether this age-associated decline is caused by reduced transcription of the pri-miRNA, a change in miRNA processing, or increased degradation, we quantified pri-miR-181a/b transcripts. pri-miR-181a/b transcripts in humans are encoded by two genes, one on Chr1q32.1 and one on Chr9q33.3, with the former predominantly expressed in T cells[28,33]. Pri-miR-181a/b1 transcripts were significantly reduced in naïve CD4 T cells from older compared to younger individuals suggesting a transcriptional mechanism in the loss of miR-181a (Fig. 1a). To determine whether chromatin accessibility at the *pri-miR-181a/b1* locus changes with age, we performed ATAC-sequencing of naïve CD4 T cells. Results in Fig. 1b show that the transcription start sites (TSS) are equally open in young and old CD4 T cells. Downstream of the TSS, two intronic sequence stretches between 198,904 and 198,905 kb are accessible, again with no age-associated differences. In reporter gene assays, the sequence at the TSS displayed promoter activity (Fig. 1c) that did not change with age (Fig. 1d). The sequence from 198,904,065 to 4558, identified as Peak 1 in Fig. 1b, was able to induce luciferase activity when cloned upstream of the minimal promoter in the pGL4.27 vector, consistent with the interpretation that this region functions as an enhancer (Fig. 1e). In contrast, the isolated Peak 2 sequence did not confer any activity. When transfected into naïve CD4 T cells, Peak 1 sequence conferred enhancer activity only in cells from young but not from older individuals (Fig. 1f). Chromatin immunoprecipitation-polymerase chain reaction (ChIP-PCR) of the presumptive enhancer sequence confirmed binding of p300, acetylated H3K27, and mono-methylated H3K4 in naïve CD4 T cells from young individuals consistent with active enhancer function (Fig. 1g). In contrast, no significant binding of acetylated H3K27 or p300 was found in older individuals.

**Regulation of the *pri-miR-181a/b1* enhancer by MYB and YY1**. The presumptive enhancer Peak 1 sequence was analyzed for TF-binding motifs using PROMO and TRANSFAC identifying YY1 and MYB as candidates that may control enhancer activity. HEK293T cells were transfected with small interfering RNA (siRNA) specific for these TFs; IKAROS or NFAT siRNA were included as irrelevant control TFs. Reporter gene assays with the construct including Peak 1 sequences were performed to

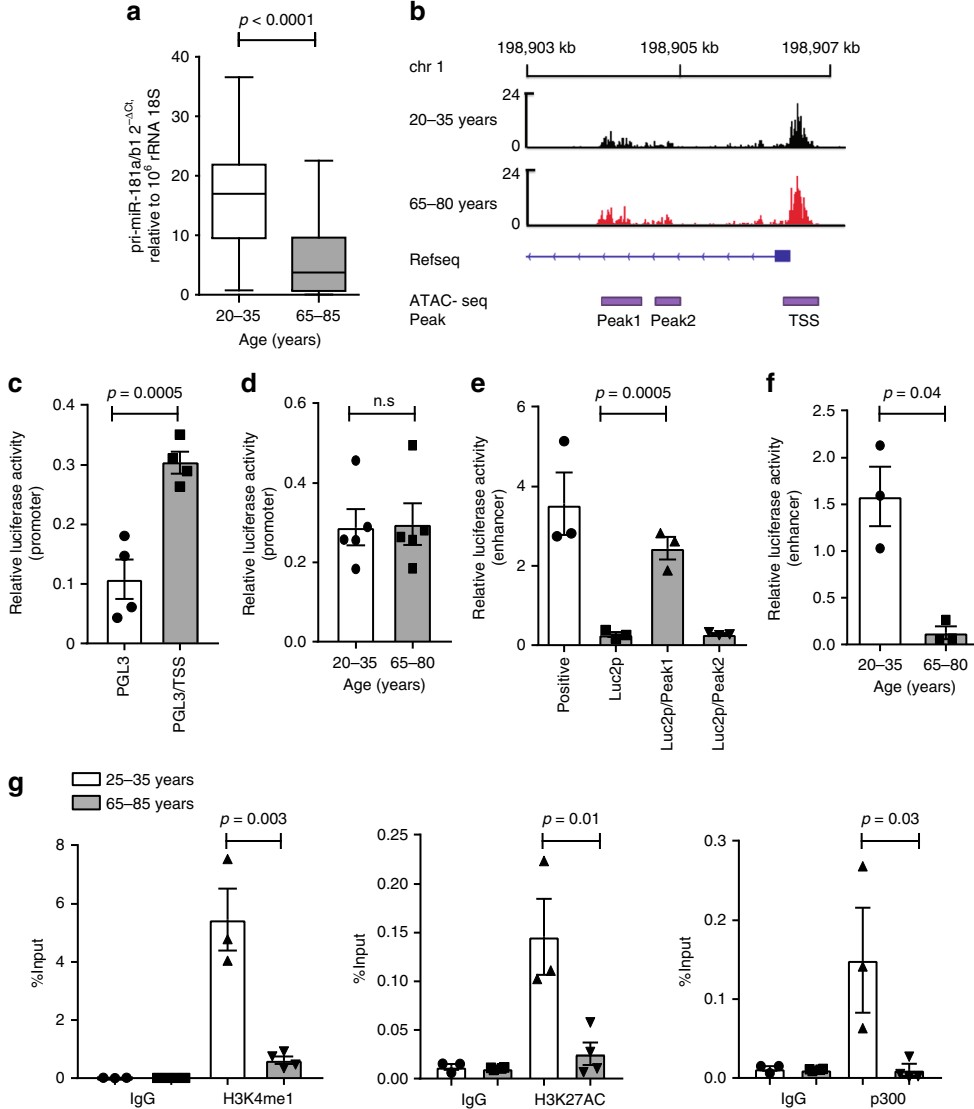

**Fig. 1** Age-associated loss of miR-181a in naïve CD4 T cells with age is caused by reduced enhancer activity. **a** Expression of pri-miR-181a/b1 in naïve CD4 T cells was compared in 20–35 ($n = 20$) and 65–85-year-old individuals ($n = 23$). Results are shown as box plots of $2^{-\Delta CT}$ (pri-miR-181a/b1 transcripts normalized to $10^6$ 18s rRNA) with medians, 25th and 75th percentile as boxes and 10th and 90th as whiskers. **b** Regions of chromatin accessibility were determined by ATAC-seq of naïve CD4 T cells. Tracks shown are merged from six young and four old adults and show open regions at 198,904,065–198,904,558 (Peak 1), 198,904,881–198,905,130 (Peak 2) and adjacent to the TSS of *pri-miR-181a/b1* on chromosome 1 irrespective of age. **c** Dual luciferase reporter assays using pGL3-basic plasmids confirms promoter activity of the sequence adjacent to the TSS of miR-181a/b1 shown as mean ± SEM ($n = 4$). **d** Results from reporter gene assays using the *pri-miR-181a/b1* promoter constructs transfected into naïve CD4 T cells are shown as mean ± SEM from young ($n = 5$) and old adults ($n = 5$). **e** Sequences corresponding to Peak 1 and Peak 2 were cloned into a pGL4.27 [luc2P/minP/Hygro] plasmid. Dual luciferase reporter assays showed enhancer activity for Peak 1 but not Peak 2 sequences; mean ± SEM are shown ($n = 3$). **f** Dual luciferase reporter assays showed increased enhancer activity for young naïve CD4 T cells. Results are shown as mean ± SEM from young ($n = 3$) and old ($n = 3$) adults. **g** ChIP-PCR for Peak 1 sequences were performed with antibodies to H3K4me1 (left), H3K27AC (middle), and p300 (right). Results are shown as mean ± SEM of naïve CD4 T cells from young ($n = 3$) and old ($n = 4$) adults. All comparisons were done by two-tailed unpaired *t* test

determine whether reduced expression of the candidate TF affected its activity. Results shown in Fig. 2a suggest that YY1 and MYB are regulators of the Peak 1 enhancer, while IKAROS and NFAT are not involved. Conversely, overexpression of YY1 or MYB increased enhancer activity in reporter gene assays in HEK293T cells (Fig. 2b, c). Silencing of YY1 reduced transcription of pri-miR-181a/b1 in HEK293T cells by more than 50% (Fig. 2d). Similar results on pri-miR-181a/b1 transcription were obtained with primary CD4 T cells in which YY1 was overexpressed (Fig. 2e) or silenced (Fig. 2f). Results were

reproducible with a second YY1 siRNA targeting a different sequence (Fig. 2f).

The Peak 1 sequence has two sequence motifs that could account for YY1 binding (Fig. 2g). ChIP-PCR with YY1 antibodies yielded signals for sequences encompassing either motif (Fig. 2h). In reporter gene assays of constructs with YY1 motifs mutated as indicated in Fig. 2g, mutation of one binding site alone only had a small effect on the reporter activity; a larger effect was seen when both sites were mutated (Fig. 2i).

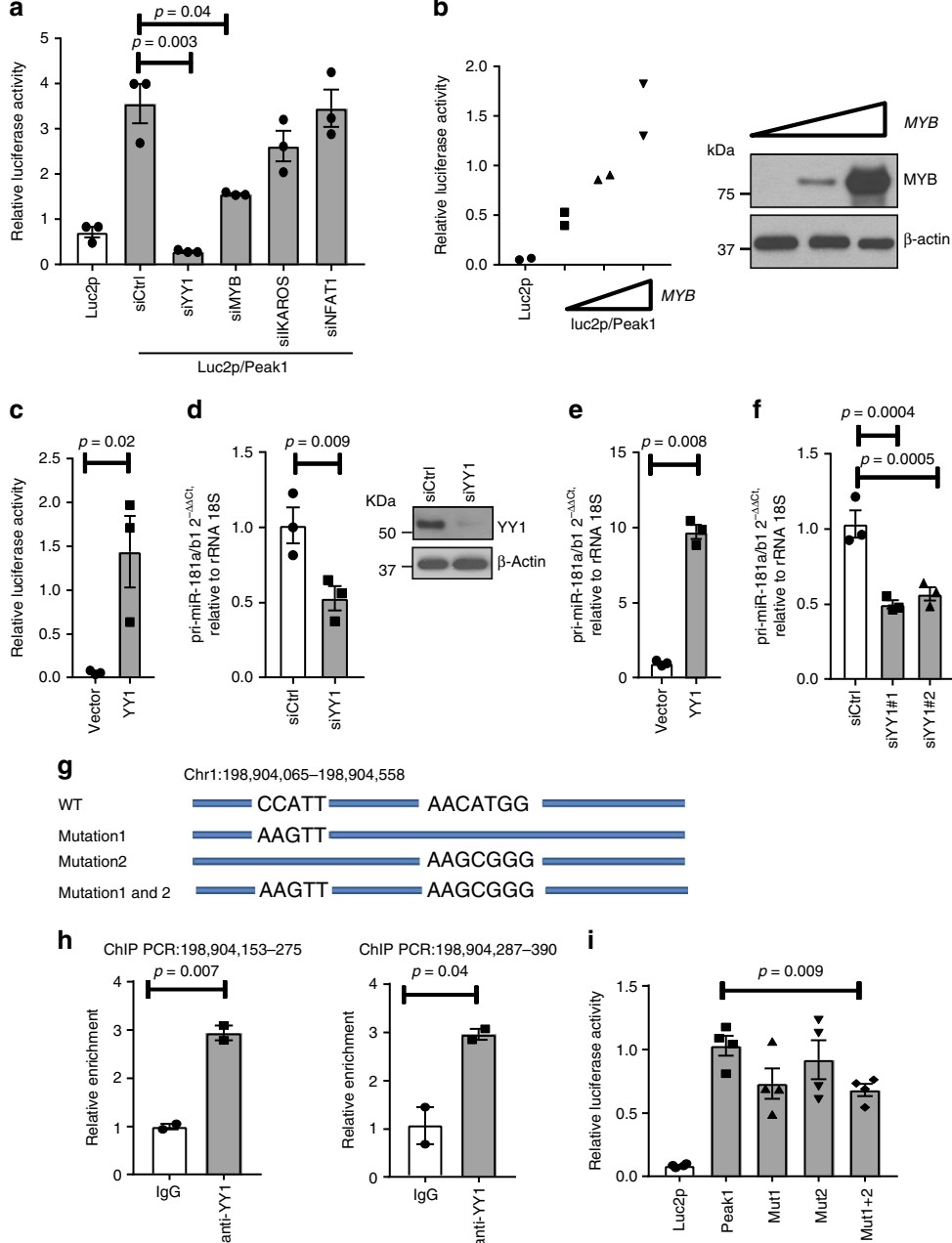

**Fig. 2** Transcription factor (TF) networks regulating the *pri-miR-181a/b1* enhancer. **a** siRNAs to TFs with binding motifs in the Peak 1 sequence (YY1, MYB) or to unrelated TFs (IKAROS, NFAT1) were co-transfected with the Luc2p/Peak 1 vector into HEK293T cells. Forty-eight hours later, enhancer activity was tested by the dual luciferase reporter assay. Results are shown as mean ± SEM ($n = 3$). **b** MYB-expressing or empty plasmid were co-transfected with Luc2p/Peak 1 reporter into HEK293T cells. Reporter activities after 48 h are shown as mean ± SEM (left, $n = 2$). Western blots confirmed MYB overexpression (right). **c** HEK293T cells were transfected with a plasmid expressing YY1 or an empty plasmid together with the Luc2p/Peak 1 reporter. Enhancer activity assessed after 48 h is given as mean ± SEM of triplicates. **d** siRNA targeting YY1 (YY1 siRNA#1) or control siRNA were transfected into HEK293T cells. pri-miR-181a/b1 transcripts quantified by qPCR are shown as mean ± SEM of triplicates (left). Western blots in the right panel document knockdown efficiency. **e, f** Naïve CD4 T cells were transfected with YY1 plasmid (**e**) or YY1 siRNA#1 and siRNA#2 (**f**), and pri-miR-181a was quantified after 48 h by qPCR. Results are shown as mean ± SEM ($n = 3$). **g** Potential YY1 binding sites at the *pri-miR-181a/b1* enhancer region. Wild-type sequences as well as mutated regions used for reporter gene assays in (**i**) are shown. **h** ChIP assays of Jurkat T cells using anti-YY1 antibodies and primer sets amplifying the indicated two potential binding sites are shown. Results shown are representative of two experiments. **i** Reporter gene assays using mutated variants of the YY1-binding site as indicated in (**g**). Results are shown as mean ± SEM ($n = 4$). Comparisons in **a**, **c**, **e**, **h**, and **i** were done by two-tailed unpaired *t* test, **d**, **f** by two-tailed paired *t* test

**Influence of age on YY1 expression**. To determine whether a change in expression of YY1 and/or MYB accounts for the loss of miR-181a with age, we compared their transcripts in naïve CD4 T cells from sixteen 20–35-year-old and eighteen 65–85-year-old individuals by quantitative PCR (qPCR). No difference was seen for *MYB* transcription (Fig. 3a); in contrast, the expression of *YY1* transcripts significantly declined with age ($p = 0.005$, Fig. 3b). Reduced YY1 expression in naïve CD4 T cells from older individuals was also seen at the protein level as determined by Western blotting (Fig. 3c, $p = 0.0007$). Moreover, a significant

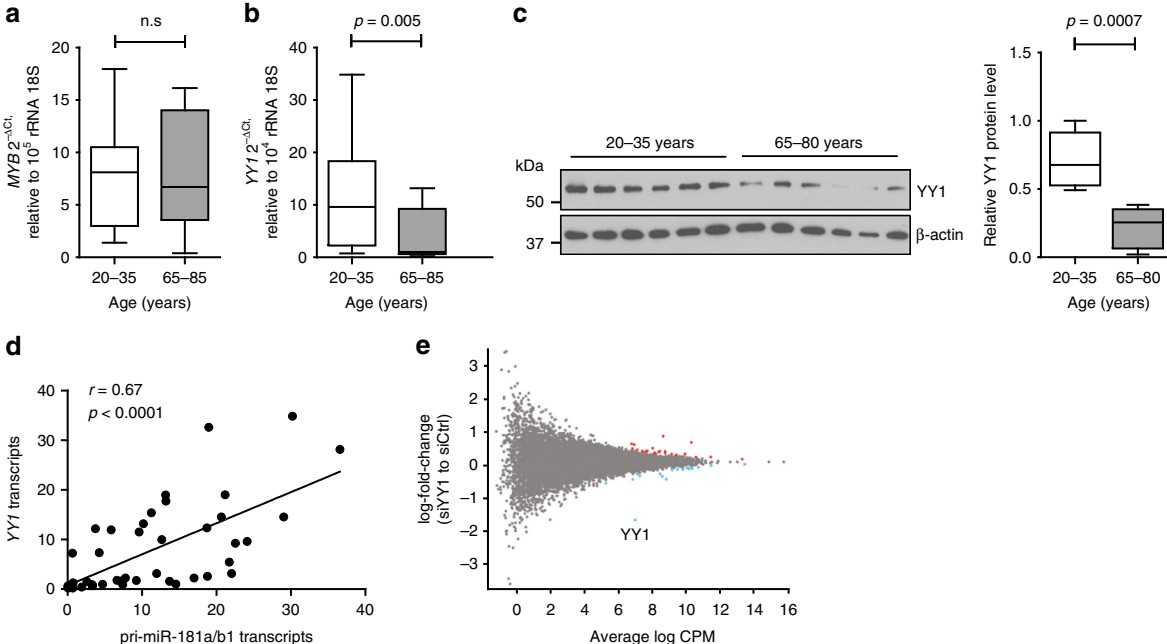

**Fig. 3** Age-associated loss in YY1 expression accounts for reduced pri-miR-181a/b1 expression. **a** *MYB* transcripts in naïve CD4 T cells were quantified by qPCR. Results for 20–35-year-old ($n = 16$) and 65–85-year-old ($n = 18$) individuals are shown as box plots of *MYB* transcripts relative to 18s rRNA. Comparison was done by two-tailed unpaired *t* test. **b** *YY1* transcripts in naïve CD4 T cells from young ($n = 21$) and old ($n = 23$) adults are compared by two-tailed unpaired *t* test. **c** YY1 protein expression in naïve CD4 T cells from young and older adults are compared by two-tailed unpaired *t* test. Representative Western blots (left) and summary statistics of YY1 protein relative to β-actin from young ($n = 9$) and older ($n = 9$) adults (right). **d** *YY1* transcripts are correlated with pri-miR-181a/b1 expression using two-tailed Pearson's correlation. **e** MA plot of RNA-seq data comparing transcriptome of young ($n = 3$) naïve CD4 T cells 48 h after transfection with control siRNA and *YY1* siRNA #1. Significant upregulation by YY1 silencing is indicated by red, and downregulation by blue using an adjusted *p* value cutoff of <0.05

correlation between *YY1* and pri-miR-181a//b1 transcripts was observed ($r = 0.67$, $p < 0.0001$, Fig. 3d). To determine whether such a close relationship between YY1 protein concentration and transcription extends to other genes, we performed RNA-sequencing on naïve CD4 T cells from three individuals that had been transfected with control or YY1 siRNA. Cells were harvested 48 h after transfection at a time point when YY1 protein concentrations had declined. As shown in the MA plot, *YY1* transcripts were the only transcript that were highly reduced in expression (Fig. 3e). Changes in other transcripts were small and only reached significance for 39 transcripts (Supplemental Table 1).

**Regulation of miR-181a targets by YY1.** In our previous studies, we have shown that age-associated loss of miR-181a impairs TCR signaling by upregulating DUSP6[34]. One other important signaling molecule targeted by miR-181a is SIRT1, which also increases with age in naïve CD4 T cells. Overexpression of pre-miR-181a reduced the protein level of DUSP6 as well as SIRT1 (Fig. 4a). Consistent with YY1 expression controlling the expression of miR-181a, we saw upregulation of DUSP6 and SIRT1 in YY1-silenced cells (Fig. 4b, c). Conversely, over-expression of YY1 or pre-miR-181a reduced DUSP6 and SIRT1 expression (Fig. 4d, e). The increased DUSP6 and SIRT1 expression due to YY1 silencing was reversed by overexpressing pre-miR-181a, consistent with the model that YY1 regulates DUSP6 and SIRT1 expression through miR-181a (Fig. 4f, g).

**YY1 regulates TCR signaling through miR-181a.** miR-181a has been shown to control the TCR activation threshold through several phosphatases, in particular DUSP6. Consistent with the specificity of DUSP6 for pERK, we found a preferential reduction in ERK phosphorylation after TCR stimulation with age[34]. This

effect can be reproduced by YY1 silencing that blunted the ERK response after CD3/CD28 cross-linking and that could be reversed by overexpressing pre-miR-181a (Fig. 5a). Restoration of the ERK phosphorylation was also accomplished by silencing of DUSP6 (Fig. 5b), but not of SIRT1 (Fig. 5c), consistent with the different function of these two enzymes in the TCR signaling cascade.

To analyze the functional consequences of YY1 expression in naïve CD4 T cells, we studied the induction of the activation marker CD69 and the production of interleukin-2 (IL-2) after TCR activation. Silencing of YY1 reduced the activation-induced expression of CD69 (Fig. 6a). Induction of CD69 could be restored by co-transfecting pre-miR-181a or DUSP6 siRNA, while silencing of SIRT1 had a minimal effect, consistent with the notion that CD69 expression is a sensitive marker of MAPK activation. YY1 silencing also negatively affected production of IL-2, again reversible by overexpressing pre-miR-181a (Fig. 6b). Here, DUSP6 and SIRT1 overexpression both appear to contribute because there was a trend towards partial restoration by DUSP6 as well as SIRT1 silencing.

**Discussion**

miR-181a is one of the most abundant microRNAs in lympho-cytes where it is dynamically regulated with T cell activation and differentiation, suggesting that it is an important determinant of the T cell differentiation state and associated functions. Of particular interest, expression of miR-181a declines with age, accounting for some of the functional changes in T cell aging[34]. Here, we tracked back the dynamic expression of miR-181a to the TF YY1 that controls a putative enhancer region of the *pri-miR-181a/b1* gene. Moreover, YY1 expression declined in naïve CD4 T cells with age and was lower in more differentiated T cells such as memory T cells (Supplementary Fig. 1), mirroring the

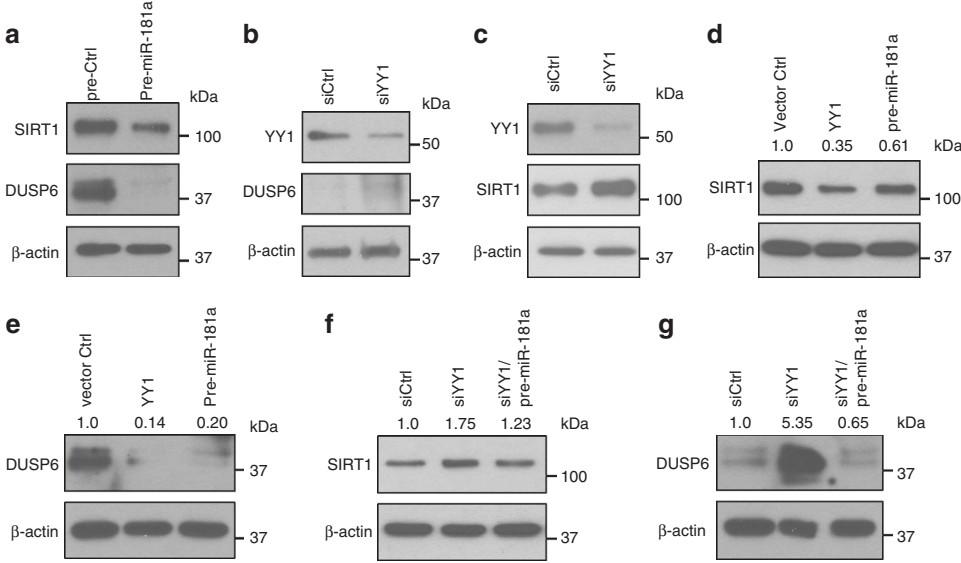

**Fig. 4** Loss of YY1 accounts for increased expression of miR-181a targets. **a** Overexpression of pre-miR-181a down-regulates SIRT1 and DUSP6 expression in CD4 T cells. Immunoblots of SIRT1 and DUSP6 after transfection with pre-miR-181a for 48 h. **b**, **c** Knockdown of YY1 enhances the expression of DUSP6 (**b**) and SIRT1 (**c**) in CD4 T cells. Representative immunoblots with YY1 siRNA #1 are shown. **d**, **e** SIRT1 (**d**) and DUSP6 expression (**e**) after transfection with YY1-expressing plasmids and pre-miR-181a. **f**, **g** Immunoblot analysis of SIRT1 (**f**) and DUSP6 (**g**) in naïve CD4 T cells transfected with control siRNA, siYY1 alone, or siYY1 combined with pre-miR-181a. All immunoblots are representative of two to three experiments with comparable results for the two YY1 siRNA

expression of miR-181a. These data suggest that the regulation of miR-181a expression in T cell differentiation as well as T cell aging is related to concentration changes in YY1 expression.

YY1 is a DNA-binding protein, which can induce or repress transcription by recruiting coactivators and corepressors that frequently do not have sequence-specific DNA-binding domains[35–37]. It is constitutively expressed and can undergo a large number of post-translational modifications that influence its function[37–42]. It is known to interact with many proteins involved in transcriptional regulation such as TATA-binding protein TFIIB, c-MYC, SP-1, ATF/CREB, and NF-κB. Through its REPO domain, YY1 can recruit Polycomb group proteins inducing gene repression through histone methylation[43–46]. YY1's ability to mediate chromosomal looping is most evident in immunoglobulin heavy chain rearrangement, where it facilitates locus contraction to facilitate V–D–J recombination[47,48].

In our previous studies of the impact of age on the epigenome of CD8 T cells, genome-wide reduced accessibility to YY1 sites was one of the most prominent findings of aging, only second to reduced NRF1 accessibility[19]. In the vicinity of the *miR-181a/b1* gene, we did not find any differences in chromatin accessibility and even the implicated enhancer region was fully accessible with age or differentiation (Fig. 2b). However, ChIP assays showed increased H3K27 acetylation and p300 occupancy at the miR-181a enhancer in T cells from young individuals, consistent with the interpretation that YY1 functions by recruiting p300 with relatively small changes in YY1 concentrations having a large impact. Recruitment of additional TFs or coactivators may also depend on YY1 concentrations to regulate miR-181a expression. So far, it is unknown whether YY1 can directly bind with MYB that regulated the miR-181a reporter but did not change with age in protein expression. However, a cooperative activity of YY1 and MYB has recently been shown in the transcriptional regulation of another microRNA, miR-155[49].

YY1 is essential for normal development, constitutive ablation is associated with peri-implantation lethality and chimeric mice expressing varying amounts of YY1 exhibit impaired embryonic growth and viability in a dose-dependent manner[50–52]. Genome-

wide expression profiling of embryonic fibroblasts from such mice identified YY1 target genes that have been implicated in cell growth, proliferation, cytokinesis, apoptosis, development, and differentiation. Such a concentration dependency was also the case for pri-miR-181a/b1 transcription where YY1 overexpression or partial silencing modified miR-181a expression. We therefore expected that the reduced YY1 expression in older individuals has more far-reaching consequences for T cell aging than only those mediated by target genes of miR-181a. However, YY1 silencing in quiescent naïve CD4 T cells only had minor effects on the transcriptome, with very few changes reaching significance, suggesting that under steady-state conditions YY1 concentrations are not important for basic cellular maintenance. This conclusion is consistent with studies of T cell-specific YY1-deficient mice, in which global T cell defects were not observed[53].

In contrast to the minor changes in the transcriptome, YY1 silencing had a clear effect on T cell activation with defects in TCR signaling events and the expression of activation markers. These defects were largely reversible by overexpression of pre-miR-181a or silencing of DUSP6 and SIRT1, known targets of miR-181a. Why regulation of miR-181a appears to be more sensitive to changes in YY1 concentrations than other YY1 target genes is unresolved. We analyzed the sequence of the putative pri-miR-181a enhancer region for sequence motifs that could indicate a unique cooperative interaction of TFs, such as between MYB and YY1, but did not find any evidence for such a model.

Obviously, the absence of a major defect under steady conditions does not preclude that YY1 concentrations are important for T cell differentiation and that YY1 silencing has a larger impact on the transcriptome in activated and differentiating T cells. Indeed, the transcriptome study in YY1-deficient embryonic fibroblasts identified target genes related to cell cycling[52], none of which showed up in our transcriptome analysis of quiescent naïve CD4 T cells. Moreover, a recent study examining genome-wide histone modifications, open chromatin and gene expression of naive, terminal-effector, memory-precursor and memory CD8+ T cell populations induced during the in vivo response to bacterial infection identified YY1 as the top-ranked

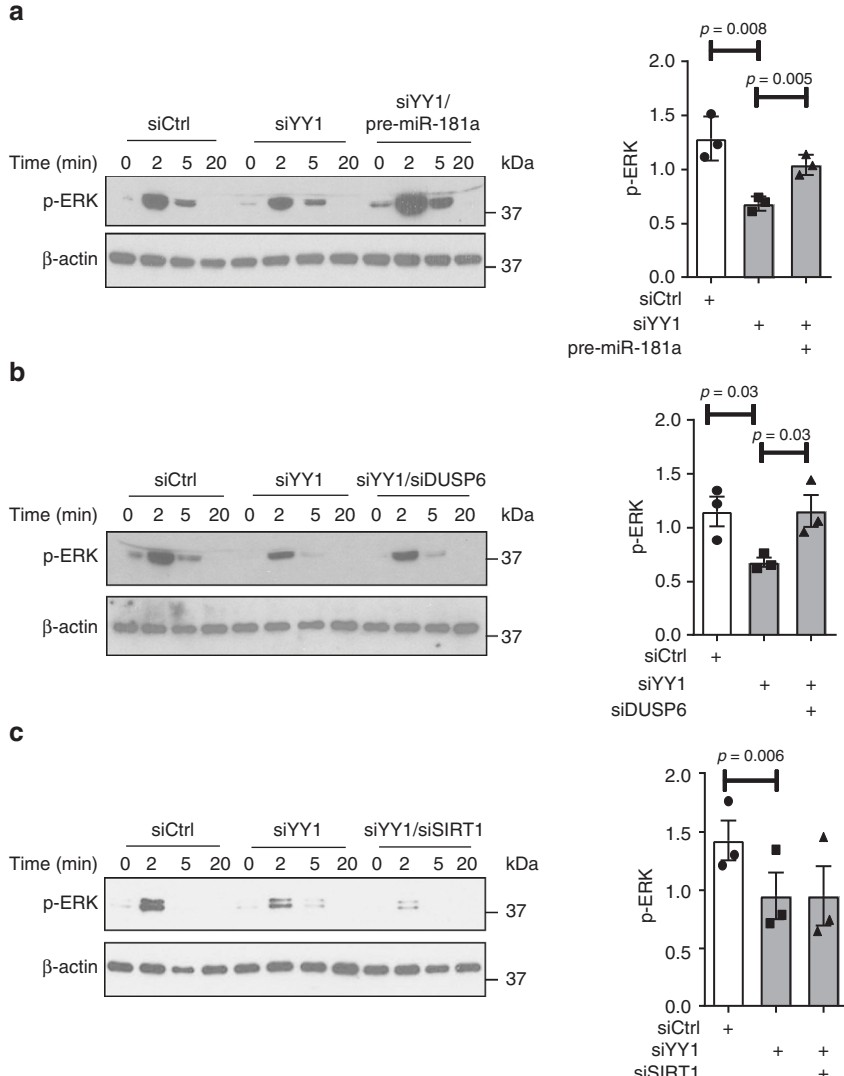

**Fig. 5** YY1 regulates T cell receptor signaling through controlling miR-181a expression. Immunoblot analysis of pERK (Thr202/Tyr204) upon CD3/CD28 cross-linking of naive CD4 T cells. Cells were transfected with control siRNA, YY1 siRNA alone (#1 or #2) or YY1 siRNA combined with either pre-miR-181a (**a**), siDUSP6 (**b**) or siSIRT1 (**c**). Results are shown as representative immunoblots (left) and mean ± SEM of pERK relative to β-actin levels at 2 min from one experiment with YY1 siRNA#1 and two experiments with YY1 siRNA#2 (right). All comparisons were done by paired $t$ test ($n = 3$)

TF in CD8 effector T cell differentiation[54]. In TH1 responses, YY1 has been shown to increase interferon-γ (IFN-γ) production, possibly by interacting with two adjacent NFAT binding sites in the *IFNG* promoter[55]. In TH2 differentiation, YY1 influences GATA3 binding and mediates chromatin remodeling and chromosomal looping of the Th2 cytokine locus to regulate Th2 cytokine genes[53]. Conversely, in Treg differentiation YY1 inhibits SMAD3/4 binding and chromatin remodeling of the FOXP3 locus[56]. It is unclear how far these studies can be extrapolated to human T cell aging, in particular, because these murine T cells completely lack YY1 expression. It remains to be explored whether reduced YY1 expression in activated T cells from older individuals accounts for some of the age-associated changes that are seen with T cell expansion and differentiation.

In our current studies of T cell activation, the functional consequences of reduced YY1 expression could be entirely attributed to miR-181a. miR-181a expression is an important regulator of T cell function. Most widely studied is its role as an intrinsic regulator of TCR signaling thresholds by targeting several phosphatases that function as negative regulators of key signaling molecules including CD3ζ and ERK. In the thymus,

miR-181a is dynamically regulated during positive and negative selection, with high expression facilitating positive selection[31]. Conversely, low expression with age appears to account for a reduced TCR ability to respond to stimuli[34,57]. However, the effects of miR-181a expression on T cell activation and differentiation are more complex than only regulating TCR signaling strength. Although DUSP6 silencing biases towards TH1 differentiation in vitro, presumably by increasing signaling strength, deletion of *miR-181a/b1* in mice favored the induction of a TH1 response[58,59]. Moreover, clonal expansion of activated T cells in miR-181a/b1-deficient mice is increased rather than decreased[33]. These findings may reflect other targets of miR-181a that are important in signal propagation. By targeting PTEN, miR-181a can modify PI3K-AKT activation[32,60]. The resulting reduced metabolic activity in miR-181a-deficient mice would decrease rather than increase clonal expansion. More important may be the targeting of SIRT1 that is increased in miR-181a-deficient mice as well as T cells from older individuals and leads to increased deacetylation[29]. IL-2 production reduced in YY1-deficient T cells could be in part restored by silencing SIRT1. Also, miR-181a is a negative regulator of key NF-κB signaling

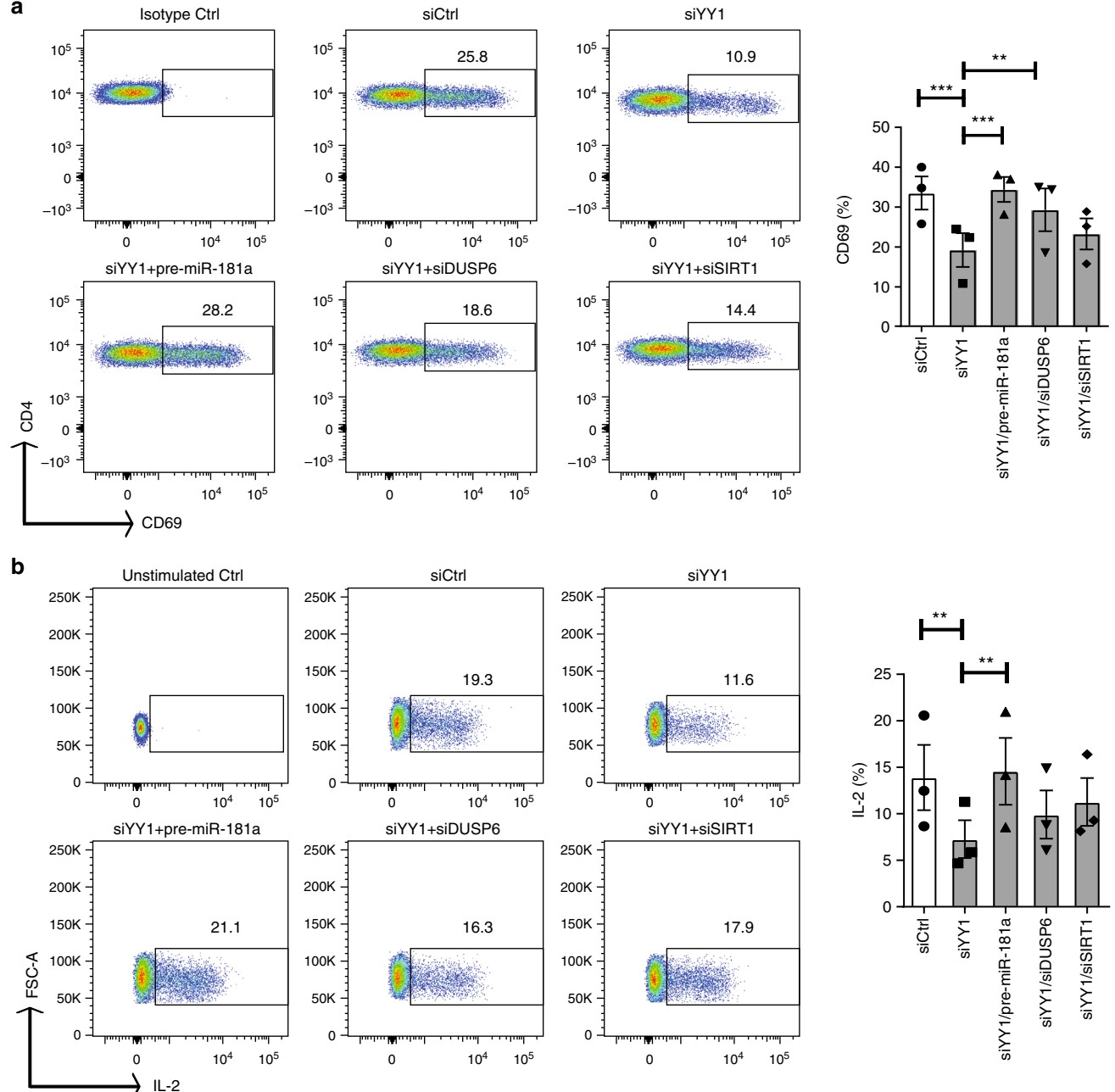

**Fig. 6** YY1 regulates T cell activation through controlling miR-181a expression. CD4 naïve T cells were transfected with control siRNA, YY1 siRNA (#1 or #2), a combination of YY1 siRNA and pre-miR-181a, a combination of YY1 siRNA and DUSP6 siRNA or a combination of YY1 siRNA and SIRT1 siRNA. **a** Cells were cultured in plates coated with 1 µg CD3/CD28 and analyzed for CD69 expression by flow cytometry after 24 h. Results are shown as scatter plots (left) and as mean ± SEM percentages of CD69$^+$ cells from one experiment with YY1 siRNA#1 and two experiments with YY1 siRNA#2 (right). **b** Transfected CD4 T cells were stimulated with CD3/CD28 Dynabeads (at a cell to beads ratio of 3:1) for 24 h, then restimulated with ionomycin and PMA in the presence of Golgi blocker and assayed for the production of IL-2. Results are shown as representative scatter plots (left) and bar graphs of mean ± SEM from one experiment with YY1 siRNA#1 and two experiments with YY1 siRNA#2. One-way ANOVA and pairwise comparison using the Tukey's method were used in all experiments. **P < 0.005. ***P < 0.0001

components, thereby slowing proliferation and tumor progression in diffuse large B cell lymphomas[61]. Further studies are needed to understand how these different targets synergize and to develop predictive models of T cell responses in older individuals low in miR-181a expression.

## Methods

**Study population**. Healthy individuals aged 20–35 years ($n = 104$) or 65–85 years ($n = 102$) with no evidence of acute illness, current or previous history of immune-mediated diseases, cancer except limited basal cell carcinoma, or any chronic disease not controlled on oral medications were included. The study was approved by the Stanford University Institutional Review Board, and all participants gave written informed consent. Peripheral blood mononuclear cells (PBMCs) were separated by Ficoll (STEMCELL Technologies) from venous blood. CD4 naïve T cells were purified from PBMC by negative selection with human CD4 T cell enrichment cocktail (15062, STEMCELL Technologies) followed by negative selection using anti-human CD45RO microbeads (130-046-001, Miltenyi Biotec). Subset purity monitored by FACS routinely exceeded 95%.

**Antibodies**. Antibodies specific for YY1 (ab12132), p300 (ab10485), histone H3 mono-methylated K4 (ab8895), and DUSP6 (ab76310) from Abcam, anti-acetyl-histone H3 (Lys27) antibodies from EMD Millipore (07–360), and antibodies against pERK (T202/Y204, 4377s), SIRT1 (2310s), and MYB (12319S) from Cell Signaling Technology were used at a dilution of 1 in 1000 for immunoblotting or 1 in 50 for ChIP. β-Actin-specific antibodies were from Sigma-Aldrich (A5441, used at 1 in 5000). Fluorochrome-conjugated antibodies were from BD Biosciences (CD4-V450 (560345), CD8-V500 (560774), CD45RA-AF700 (560673), and CD69-PerCP-Cy5.5 (560738)) and BioLegend (CD62L-PE (304806), and human IL-2 Alexa-Fluor-488 (500314)). All antibodies for flow cytometry were used at 1 in 50. Flow cytometry gating strategies are included in Supplementary Fig. 5.

**ATAC-sequencing**. ATAC-sequencing (ATAC-seq) on naïve CD4 T cells purified by cell sorting was performed using standard protocols as described[19]. Data were aligned to hg19 using Bowtie 2. Gene and TSS annotations were drawn from RefSeq and GENCODE.

**RNA-sequencing**. Naïve CD4 T cells were isolated from PBMC using EasySep Human Naïve CD4$^+$ T cell Enrichment Kit (STEMCELL Technologies, Cat. # 19555). One million cells were transfected with control siRNA (Dharmacon, D-001910-01) or YY1 siRNA (Dharmacon, A-011796-16) using the P3 Primary cell 4D-Nucleofector Kit and the Lonza 4D-Nucleofector System (Lonza). Transfected cells were cultured in RPMI-1640 media supplemented with 10% fetal bovine serum (FBS) for 48 h until collection. Libraries were prepared using the NuGEN Ovation Whole Blood Kit according to the manufacturer's instructions and sequenced on an Illumina 2500 HiSeq.

RNA-sequencing (RNA-seq) reads were aligned to hg19 using STAR 2.5.3a aligner[62], followed by generation of transcript-level counts using featureCounts[63] package. Normalization and identification of differentially enriched genes was performed using edgeR[64,65] and limma packages from R Bioconductor and modeled to identify differences in paired samples in control and YY1-knockdown groups.

**Reporter constructs and luciferase reporter assays**. Accessible regions within the pri-miR-181a/b1 gene, as identified by ATAC-seq from young and old naïve CD4 T cells, were cloned into reporter constructs as follows. The region upstream of the TSS (chr1: 198,906,516–198,906,936) was amplified with forward primer: 5′-GGGGCTCGAGTTAGGTTGAATAGAATTCCCA-3′ and reverse primer: 5′-GGGAAGCTTTGATAGGATGAGCAAACAAA-3′ and cloned into the pGL3-basic plasmid (Promega). Peak 1 sequences, chr1: 198,904,065–198,904,558 and Peak 2: chr1: 198,904,881–198,905,130, of pri-miR-181a/b1 were amplified by PCR using Peak 1 primers (forward: 5′-GGGCTCGAGCTCTTTATGAGAATATTTA CATT-3′ and reverse: 5′-GGGAGATCTATTGTTTTTCTTTTCCCATGTT-3′) and Peak 2 primers (forward: 5′-GGGCTCGAGTGGGAACACATAATAAGATA-3′ and reverse: 5′-GGGAGATCTATATTGGGATCAGAGAGTTC-3′). Purified PCR products were cloned into the pGL4.27 Luc2p/minp vector (Promega). YY1 binding sites mutant were generated using Site-Directed Mutagenesis Kit (#200523, Agilent Technologies) and the following primers: mutant 1 (forward: 5′-CAACT TATTTTTGTGGGAAGTTAGATTTCAGAATTCAC-3′; reverse: 5′-GTGAA TTCTGAAATCTAACTTCCCACAAAAATAAGTTG-3′) and mutant 2 (forward: 5′-CCAGATTTGTACAAgcGgGGACGGAATCATCAATAT-3′; reverse: 5′-ATA TTGATGATTCCGTCCcCgcTTGTACAAATCTGG-3′); mutant 1 + 2 was generated with mutant 1 as template and mutant 2 primers. All inserts were verified by sequencing.

Thirty nanograms of pri-miR-181a/b1 Luc2p/Peak 1, Luc2p/Peak 2, or Luc2p empty vector (negative control) or 30 ng pri-miR-181a/b1 pGL3/TSS or pGL3-basic control plasmids (negative control) was co-transfected with 1 ng Renilla luciferase reporter pRL-TK (Promega) into HEK293T cells. Forty-eight hours later, cells were collected and enhancer or promoter activity was determined using the Dual Luciferase Reporter Assay System (Promega, E1910).

Candidate transcript factors predicted to bind to Peak 1 and 2 sequences were identified using PROMO and TRANSFAC. To assess their functional activity in regulating pri-miR-181a/b1 expression, reporter gene assays were performed with HEK293T cells after transfection with siRNAs from Dharmacon targeting MYB (M-003910-00), IKAROS (L-019092-02), YY1 (A-011796-16), and NFAT1 (A-003606-15). Non-targeting siRNA (D-001910-01-05) was used as a negative control. To compare promoter and enhancer activity in primary cells from young and older individuals, 3 μg pri-miR-181a/b1 Luc2p/Peak 1 firefly and 500 ng Renilla luciferase reporter plasmids were transfected into naive CD4 T cells with two different YY1 siRNA (Dharmacon, #1 A-011796-16 and #2 A-011796-17). Reporter activity was determined after 48 h.

**ChIP assay**. ChIP assays were performed on five million CD4-naïve T cells isolated from 20–35-year-old and 65–85-year-old individuals using the ChIP-IT Kit (53040) from Active Motif. Oligonucleotide primers were designed to amplify pri-miR-181a/b1 sequences: chr1/198,904,153–275, set 1—forward: 5′-GTGGGC CATTAGATTTCAGAA-3′ and reverse: 5′-CAAATCACAGCAGCTTCTATC-3′; chr1: 198,904,287–390, set 2—forward: 5′-TTGTCTTTTTACCCCCTCGT-3′ and

reverse: 5′-TGATTCCGTCCATGTTTGTAC-3′. Genomic DNA from CD4-naïve T cells was used as input control.

**Quantitative PCR**. Total RNA was extracted with the RNeasy Plus Micro Kit (Qiagen, 74034), and cDNA was synthesized using the High-Capacity RNA-to-cDNA Kit (Applied Biosystems, 4387406). qPCR was performed in duplicates in 384-well plates using the ABI 7900HT System with Taqman Universal Master Mix II (Thermo Fisher) using the following probes: pri-miR-181a/b1 (Taq-man:HS03302966-pri),18S rRNA (Taqman:Hs99999901_s1), YY1 (Taqman: HS00231533-m1), SIRT1 (Taqman:HS01009006-m1), MYB (Taqman:hS00920556-m1), and ACTB (Taqman:HS99999903-m1).

**T cell functional assays**. CD4-naïve T cells were transfected with two different YY1 siRNA (Dharmacon, #1 A-011796-16 and #2 A-011796-17), YY1 siRNA/pre-miR-181a (Ambion, #AM17100, ID:PM10421), YY1 siRNA/siDUSP6 (Dharma-con, A-003964-14), YY1/siSIRT1 (Dharmacon, A-003540-17), and control siRNA (D-001910-01), respectively, using the P3 Primary cell 4D-Nucleofector Kit for the Lonza 4D-Nucleofector System (Lonza). Transfected cells were cultured in RPMI-1640 media supplemented with 10% FBS for 48 h and then cross-linked with 1 μg/ml CD3/CD28 antibody. ERK phosphorylation was measured by Western blot. To assess CD69 expression, cells were seeded into plates coated with 1 μg/ml anti-CD3/CD28 Ab. CD69 expression was assessed on an LSR Fortessa cell ana-lyzer (BD Biosciences) after 24 h. To assess IL-2 production, cells were cultured with anti-CD3/CD28 Dynabeads (cell to bead ratio 3:1) for 24 h before restimu-lated with 2.5 ng/ml PMA and 500 ng/ml ionomycin in the presence of Golgi plug (BD Biosciences, 555029) for additional 4 h. Cells were harvested, fixed, and per-meabilized followed by staining with the Alexa-Fluor-488-conjugated anti-human IL-2 antibodies and analyzed on the Fortessa.

**Immunoblotting**. Two million CD4 T-naïve cells from from young (aged 20–35 years) or older (aged 65–80 years) healthy donors or naïve CD4 T cells transfected with siYY1, siYY1/pre-miR-181a, siYY1/siDUSP6, siYY1/siSIRT1, or control siRNA and stimulated by anti-CD3/CD28 were lysed on ice with RIPA buffer (Thermo Scientific) supplemented with the phosphatase inhibitor sodium ortho-vanadate and the proteinase inhibitor phenylmethylsulfonyl fluoride (Santa Cruz Biotechnology). Ten micrograms total protein was separated on 4–15% gradient sodium dodecyl sulfate gels and transferred to polyvinylidene difluoride mem-branes. Membranes were incubated with primary antibodies specific for YY1, DUSP6, SIRT1, and pERK, respectively, then horse radish peroxidase-labeled secondary antibody and developed with Pierce ECL Western blotting substrate (Thermo Fisher Scientific). For loading control, the membrane was stripped with stripping buffer (Invitrogen) and re-probed with anti-β-actin antibodies. Uncropped immunoblot images are included in Supplementary Figs. 2–4.

**Statistical analysis**. Statistical analysis was performed with the Prism 5.0 software (GraphPad Software Inc.) using paired, unpaired two-tailed $t$ test or one-way analysis of variance (ANOVA) with post hoc Tukey. Sample sizes were chosen to ensure 80% power with $\alpha$ of 0.05 for detecting a difference of 1.0 standard deviation between populations. $P < 0.05$ was considered as significant. Two-way Pearson's test was performed to examine the correlation between YY1 and pri-miR-181a/b1 transcripts.

**Data availability**. The authors declare that the data supporting the findings of this study are available within the article and its Supplementary information files, or are available upon reasonable requests to the authors. RNA-seq and ATAC-seq data have been deposited in SRA with the accession codes SRP150954 and SRP151484

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

## Acknowledgements

This work was supported by the National Institutes of Health (R01 AI108891, R01 AG045779, U19 AI057266, R01 AI129191 to J.J.G. and R01 AR042527, R01 HL117913, R01 AI108906 and P01 HL129941 to C.M.W.). The content is solely the responsibility of the authors and does not necessarily represent the official views of the National Institutes of Health.

## Author contributions

Z.Y., G.L., C.M.W. and J.J.G. designed research and analyzed data. Z.Y., G.L., C.K. and B. H. performed the experimental work. R.R.J. analyzed RNA-seq data. Z.Y., C.M.W. and J. J.G. wrote the manuscript.

## Additional information

**Competing interests:** The authors declare no competing interests.

