## [Peer Review File · Nature Communications]

Reviewers' comments:

Reviewer #1 (T cell anergy/activation)(Remarks to the Author):

This is a very interesting work identifying the role of decreased YY-1 levels as contributing to decreased activity of T cells in older individuals by regulating MiR-181a. The work nicely follows up and advances previous studies concerning the role of MiR-181a in this process. Overall I enjoyed reading this paper and I think it provides important insight to the regulation of T cells during aging. The data were robust and logical. I was wondering if the authors could address the following:

1. I think the story would be strengthened if there were more data in old and young T cells looking at the expression of YY-1 target genes/proteins. Obviously the focus is on the ability of YY-1 to regulate MiR-181A but I think it would be important to better characterize the functional significance of the decreased YY-1 levels with regard to the expression of other YY-1 targets
2. Along these lines ,in the discussion (I know you do this in the last paragraph) maybe add a little more on the potential role of decreased YY-1 levels in regulating T cell activation independent of MiR-181A. Not to detract from the MiR-181a story but just to complement it.

Reviewer #2 (miRNA, Tfh, Treg)(Remarks to the Author):

In this study, Ye et al. identified YY1 as a key transcription factor that controls miR-181a expression. In particular, the authors demonstrated that in T cells isolated from older individuals, diminished transcription of pri-miR-181a is accompanied with reduced YY1 expression. Finally, the authors showed that YY1 knocked down in T cells from young individuals could lead to reduced T cell activation and function similar to what was observed in the older counterpart. While the study is generally of good quality, it is uncertain as to whether the degree of immunological advance made in this manuscript is sufficient to meet the standard for this journal.

Specific comments:

1. While YY1 was known to play a critical role in promoting Th2 cytokine production (Hwang et al. 2013), however, even in the absence of YY1, there was no other obvious T cell function defects could be detected. Therefore, I am a bit confused as how partial YY1 knocked down could lead to an even stronger T cell phenotype. It is well-known that siRNA KD could generate off-target effects. The authors should treat cells with different siRNAs. If the phenotypes are consistent among all samples treated with different shRNAs with distinct sequences yet targeting the same gene, one would feel more confident to exclude the contribution from the off-target effects.
2. Statistical analysis was lacking in several figures.

Reviewer #3 (Epigenetic, T exhaustion)(Remarks to the Author):

This study assessed the impact of differential expression of miR-181 in T cells from young vs aged individuals. Findings suggest that miR-181 expression is regulated at the level of chromatin accessibility, as ATAC-seq data identified an enhancer region that is open in young but closed in aged T cells. The authors went on to show that TFs YY1 and MYB bind at this enhancer element and are regulate miR-181 expression. The authors also identified the binding sites within YY1 and MYB that bind to the enhancer. Importantly, expression of YY1 was also found to decline with age, implicating this TF as a key regulator of the age-related decline in T cell function. Thus, overall the main strengths of the paper are the fact that while diminished miR-181a expression in aged T cells has been known to be a key factor in their functional decline, the mechanisms underlying this were

unknown. Here, the authors identify that the diminished expression of miR181a in aged T cells is a result of impaired YY1 binding to a critical enhancer. The major weakness of the paper is the loose and associative connection of the findings with DUSP6 activity. While the authors point to a role for DUSP6 in the diminished response of aged T cells to activation (ie. Page 8 "In addition to this early effect on TCR activation, likely mediated by DUSP6, YY1-mediated regulation of miR-181a/b1 also influenced T cell differentiation..."), no clear causal role for DUSP6 is identified here. This results in the conclusions not being fully supported by the data presented; at best, it becomes a distraction from the main message of the paper. Other points that should be addressed:

1. Figure 1G-I appears to be analysis of a single sample (ie cell from n=1 per group).
2. There is no statistical analysis or comparison between the groups in Figure 2.
3. The authors showed that knockdown of YY1 resulted in an increase in both DUSP6 (Figure 4a) and SIRT1 (4C). The authors go on to do more extensive analysis of SIRT1, including demonstrating that it is increased in aged vs. young T cells, but do no further analysis of the correlation between YY1 and DUSP6 in young or aged cells. The rationale for this focus is unclear.
4. Summary data of multiple replicates should be shown for the flow cytometry data in Figure 5B and 5C.
5. Figure 5 shows reduced pErk and CD69 expression in siYY1 T cells as compared to control T cells. However, in the text the authors attempt to connect these data to their previous observation of altered DUSP6 expression, when in fact there is no indication that in these cells DUSP6 is mediating these effects, since Erk phosphorylation could be altered by many other pathways. New experiments should be done to demonstrate a causal role for DUSP6 in the observed findings.
6. The IL-2 staining in Figure 5C on day 2 is not convincing. Unstimulated controls should be shown (for both day 2 and day 5), and a different type of graph should be depicted to better highlight rare events (ie dot plot, contour plot).

Reviewer #1:

1. I think the story would be strengthened if there were more data in old and young T cells looking at the expression of YY-1 target genes/proteins. Obviously the focus is on the ability of YY-1 to regulate MiR-181A but I think it would be important to better characterize the functional significance of the decreased YY-1 levels with regard to the expression of other YY-1 targets

We have performed RNA-seq in young naïve CD4 T cells transfected with siRNA for YY1 to identify target genes. Although YY1 transcripts were clearly reduced, changes in the transcriptome were minor and did not include the YY1 target genes previously identified in embryonal fibroblasts. This finding is consistent with the issue raised by Reviewer 2 that no global defects were seen in YY1-deficient T cells. Of note, all published studies that describe YY1-dependent gene expression, use systems, in which cells differentiate, but not basic cellular function under steady state conditions as in our studies. We have added Figure 3e, a supplemental table and revised the discussion.

2. Along these lines, in the discussion (I know you do this in the last paragraph) maybe add a little more on the potential role of decreased YY-1 levels in regulating T cell activation independent of MiR-181A. Not to detract from the MiR-181a story but just to complement it.

In the function analysis of T cell activation, YY1 deficiency-induced defects could be compensated by overexpressing pre-miR-181a and in part by silencing DUSP6 and/or SIRT1. These new data are included in Figures 4, 5 and 6. We considered to expand our studies to T cell differentiation, in particular given the mouse data on TH2 cells and Tregs, however, we found that such studies are difficult to control because YY1 is dynamically regulated after T cell activation.

Reviewer #2

1. While YY1 was known to play a critical role in promoting Th2 cytokine production (Hwang et al. 2013), however, even in the absence of YY1, there was no other obvious T cell function defects could be

detected. Therefore, I am a bit confused as how partial YY1 knocked down could lead to an even stronger T cell phenotype. It is well-known that siRNA KD could generate off-target effects. The authors should treat cells with different siRNAs. If the phenotypes are consistent among all samples treated with different shRNAs with distinct sequences yet targeting the same gene, one would feel more confident to exclude the contribution from the off-target effects.

As predicted by the reviewer, we did not identify major global defects in CD4 T cells under steady state conditions due to reduced YY1 expression. We used a second siYY1 targeting a different sequence as shown in Figure 2, with identical results on pre-miR-181a expression, making off-target effects unlikely. In addition, functional assays in Figures 5 and 6 were performed with both siYY1, again with similar results. Moreover, the reporter gene assays with overexpressed YY1 or mutated YY1 sequences directly implicate YY1 in regulating pri-miR-181a transcription.

2. Statistical analysis was lacking in several figures.

Statistical analysis is now included in all figures and legends.

Reviewer #3:

The major weakness of the paper is the loose and associative connection of the findings with DUSP6 activity. While the authors point to a role for DUSP6 in the diminished response of aged T cells to activation (i.e. Page 8 “In addition to this early effect on TCR activation, likely mediated by DUSP6, YY1-mediated regulation of miR-181a/b1 also influenced T cell differentiation...”), no clear causal role for DUSP6 is identified here. This results in the conclusions not being fully supported by the data presented; at best, it becomes a distraction from the main message of the paper.

1. Figure 1G-I appears to be analysis of a single sample (i.e. cell from n=1 per group).

Three ChIP assays were performed for each antibody. Results are now shown as summary graphs with statistics instead of representative examples.

2. There is no statistical analysis or comparison between the groups in Figure 2.

Statistical analysis was added.

3. The authors showed that knockdown of YY1 resulted in an increase in both DUSP6 (Figure 4a) and SIRT1 (4c). The authors go on to do more extensive analysis of SIRT1, including demonstrating that it is increased in aged vs. young T cells, but do no further analysis of the correlation between YY1 and DUSP6 in young or aged cells. The rationale for this focus is unclear.

All functional studies in Figures 4, 5 and 6 now include experiments with pre-miR-181a overexpression and DUSP6 or SIRT1 silencing.

4. Summary data of multiple replicates should be shown for the flow cytometry data in Figure 5B and 5C.

Bar graphs of mean \pm SEM are now included.

5. Figure 5 shows reduced pErk and CD69 expression in siYY1 T cells as compared to control T cells.

However, in the text the authors attempt to connect these data to their previous observation of altered DUSP6 expression, when in fact there is no indication that in these cells DUSP6 is mediating these effects, since Erk phosphorylation could be altered by many other pathways. New experiments should be done to demonstrate a causal role for DUSP6 in the observed findings.

As indicated in our response to 3, all experimental designs now include DUSP6 silencing.

6. The IL-2 staining in Figure 5C on day 2 is not convincing. Unstimulated controls should be shown (for both day 2 and day 5), and a different type of graph should be depicted to better highlight rare events (ie dot plot, contour plot).

Flow studies are now shown in Figure 6 as dot plots including negative controls as well as bar graphs summarizing experiments.

We thank the reviewers, who have helped us to improve the quality of our studies, and we are submitting the revised manuscript for your consideration.

REVIEWERS' COMMENTS:

Reviewer #1 (Remarks to the Author):

The authors have answered my questions

Reviewer #2 (Remarks to the Author):

While the authors have largely addressed my previous concern about the potential off-target effect from siRNA studies, the key question remains as to how YY1 knockdown could lead to clear T cell phenotypes (i.e. naive CD4 T cell activation and IL-2 production) especially considering that no obvious change in gene expression as shown in the RNA-seq analysis. The mere inclusion of differentially expressed genes in YY1-silenced CD4 T cells (Supp. table) is not particularly helpful.

Reviewer #3 (Remarks to the Author):

Statistical analyses are still not shown for Figure 2h.

Other than this, the authors have satisfactorily addressed all of my comments.

REVIEWERS' COMMENTS:

Reviewer #1 (Remarks to the Author):

The authors have answered my questions

Reviewer #2 (Remarks to the Author):

While the authors have largely addressed my previous concern about the potential off-target effect from siRNA studies, the key question remains as to how YY1 knockdown could lead to clear T cell phenotypes (i.e. naive CD4 T cell activation and IL-2 production) especially considering that no obvious change in gene expression as shown in the RNA-seq analysis. The mere inclusion of differentially expressed genes in YY1-silenced CD4 T cells (Supp. table) is not particularly helpful.

We agree with the reviewer that it is surprising that the YY1 silencing does not induce more striking transcriptional changes, but we feel that all data should be included and the results from the silencing experiment shown as supplemental table.

Reviewer #3 (Remarks to the Author):

Statistical analyses are still not shown for Figure 2h.

Other than this, the authors have satisfactorily addressed all of my comments.

Significance levels are provided.